# Prehospital major incident management: how do training and real-life situations relate? A qualitative study

Karin Hugelius ![ORCID] , Samuel Edelbring, Karin Blomberg

Faculty of Medicine and Health, Örebro University, Örebro, Sweden

**Correspondence to**
Dr Karin Hugelius;
Karin.hugelius@oru.se

## ABSTRACT

**Objective** To explore the relationship between preparations and real-life experiences among prehospital major incident commanders.

**Design** An explorative, qualitative design was used.

**Setting** Prehospital major incidents in Sweden. Data were collected between December 2019 and August 2020.

**Participants** Prehospital major incident commanders (n=15) with real-life experiences from major events, such as fires, bus accidents, a bridge collapse and terrorist attacks, were included. All but one had participated in 2-day training focusing on the prehospital management of major incidents. In addition, about half of the participants had participated in simulation exercises, academic courses and other training in the management of major incidents.

**Methods** Data from two-session individual interviews were analysed using inductive thematic analysis.

**Results** The conformity between real-life major incidents and preparations was good regarding prehospital major incident commanders' knowledge of the operational procedures applied in major incidents. However, the preparations did not allow for the complexities and endurance strategies required in real-life incidents. Personal preparations, such as mental preparedness or stress management, were not sufficiently covered in the preparations. To some extent, professional experience (such as training) could compensate for the lack of formal preparations.

**Conclusions** This study identified perceived gaps between preparations and real-life experiences of being a prehospital major incident commander. To minimise the gaps between demands and expectations on perceived control and to better prepare individuals for being prehospital major incident commanders, the training and other preparations should reflect complexities of real-life incidents. Preparations should develop both technical skills required, such as principles and methodology used, and personal preparedness. Personal preparations should include improving one's mental preparedness, self-knowledge and professional self-confidence required to successfully act as a prehospital incident commander. Since little is known about what pedagogical methods that should be used to enhance this, further research is needed.

## BACKGROUND

Major incidents are defined as situations where available resources are insufficient for

### Strengths and limitations of this study

► The two-session approach used for the interviews stimulated a deeper reflection and rich material to analyse.

► The experiences from many different major incidents were gathered, thus providing a broad perspective on prehospital incident management.

► The multiprofessional background within the research team added different perspectives to the interpretation of the results.

► The limited number of study participants (n=15) and the Swedish setting can impact the transferability of the results.

► Some interviews were conducted face-to-face, and others were conducted by phone. This may have influenced the sharing of experiences during the interviews.

the immediate need of medical care.[1] Moreover, these challenging situations are often characterised by high levels of uncertainty, dynamic development and time-critical decision-making.[2 3] The prehospital management of major incidents is an essential part of the overall medical response that requires specific skills and arrangements, such as an effective incident command.[4] However, structures in on-scene management vary around the world. In Sweden, at least one nurse, often specialised in prehospital emergency care, is on duty in each emergency ambulance. In the case of a major incident or disaster event, the first ambulance personnel that arrive at a scene will adopt the positions of the prehospital incident commander and medical officer in charge. These two positions may be carried out by either one or two persons. These positions ensure overall responsibility and have a mandate to lead and organise all aspects of the prehospital medical response, including overall decisions, priorities and requests, distributing prehospital resources and collaborating with relevant partners on-site. In this study, the prehospital incident commander and prehospital medical officer in charge

were studied without making any distinction between them; the term prehospital incident commander[4 5] will be used to cover both positions.

Since major incidents are relatively rare, many ambulance personnel do not have the opportunity to obtain extensive professional experience from such events. Therefore, training programmes and exercises are traditionally used to prepare responders.[6 7] Training exercises provide an opportunity to improve individual and team efforts, as well as the decision-making process, test technical equipment and working methods, and increase awareness about the complexity of disaster situations.[7 8] To ensure that ambulance personnel are sufficiently prepared to adopt the role of a prehospital major incident commander, such knowledge is essential. Despite the long history and worldwide use of educational sessions and exercises to prepare for real major incidents or disasters, little is actually known about their effects.[6 9] Furthermore, the relationship between preparations and reality is an unexplored area.[10–12]

Therefore, the aim of this study was to explore the relationship between preparations and real-life experiences among prehospital major incident commanders.

## METHODS
An explorative, qualitative design was used to gain insight into prehospital major incident commanders' experiences of major incidents in relation to previous preparations.

### Study participants and recruitment
Swedish ambulance personnel who had experience being prehospital major incident commanders and/or prehospital medical officers during a least one major incident (defined as an incident that had led to a heightened level of response within the healthcare system) during the previous 5 years were included in the study. The study participants were recruited between December 2019 and August 2020 via written invitations sent to ambulance services where major incidents had occurred during the last 5 years. In addition, the same invitation was posted on social media groups and forums aimed specifically at Swedish ambulance personnel. If interested in participating in the study, the participants were asked to access a web page for information on the study; here, the participants could sign up for the study. Subsequently, the first author (KH) contacted the participants to provide further information on the study and to set a suitable time for the first interviews. All prehospital major incident commanders who volunteered to participate in the study matched the criteria and were thus included.

### Data collection procedure
A total of 15 Swedish prehospital major incident commanders were interviewed in 27 individual interviews, conducted in two-session interviews.[13] The first interview was followed by a second interview within 2 weeks. This approach was used to ensure that reflections and thoughts that had been raised during the first interview were followed up in the second to ensure rich data.[13] Three of the study participants declined a second interview due to personal reasons (such as planned holidays). A semistructured interview guide was used, based on previous research, and had been developed in discussion by all the authors. The questions used in the first set of interviews covered demographic information, previous educational sessions and training exercises in managing major incidents and questions regarding the participants' experiences from such incidents. In the second set of interviews, which followed up on the first interview, the study participants were asked if they had any additional thoughts or experiences that had not been mentioned in the first set of interviews. The interview guide was piloted with two prospective participants, after which only small adjustments were made. Therefore, the pilot interviews were included in the analysis. In addition, all study participants were asked to complete the Swedish civilian version of the post-traumatic checklist, with a cut-off value of 40 used as an indicator for recommending professional assessment of post-traumatic stress.[14] The first and second session interviews lasted between 24 and 89 min (median 53 min) and 5 and 22 min (median 12 min), respectively. The total interview time was approximately 16 hours.

All interviews were conducted by the first author (KH), either by telephone (n=13) or face-to-face (n=2), depending on geographical distance and the COVID-19 pandemic. Face-to-face interviews were conducted in privacy at the participants' workplace. Full study information was provided before the interviews started, and digital informed consent was obtained from the participants before the first interview.

### Analysis
The interviews were transcribed in full by a professional transcriber and subsequently checked by the interviewer. An inductive thematic analysis[14] approach was used to identify the skills and experiences requested and used in real-life incidents, and to relate the preparations to those. First, all interviews were read through to gain a sense of the complete data. Features pertaining to the study purpose were then marked, extracted and coded. Initially, main themes were identified. Within these, themes and subthemes were then formed from the codes based on similarities and differences between them. Thematic maps[15] were then used to analyse the relations between the main perspectives, themes and subthemes. All themes and subthemes were labelled and described. Finally, a comprehensive summary of the results, showing the relations between the themes, was formulated. Quotations were identified and used to illustrate the findings and increase trustworthiness. The analysis was initially performed by the first author (KH), then reviewed and discussed with the other authors (SE and KB).

## Patient and public involvement

Given the focus on the experiences and training of ambulance personnel, patients and the public were not involved in the design, data collection or data analysis.

## RESULTS

Of the 15 prehospital major incident commanders that were interviewed, 10 were male and 5 were female. Their ages varied from 29 to 62 years, with a median of 38 years. Their professions were as follows: ambulance nurses with a specialisation in ambulance care (n=7), nurses with a specialisation in anaesthesia care (n=4) and general nurses (n=4). They had all been deployed within ambulance services between 1 and 39 years (median of 18 years). All except one were trained in prehospital major incidents management by participating in a standardised 2-day course consisting of both theoretical and practical training. The theoretical training covered legal aspects, reporting procedures and mass casualty triage, and the practical training consisted of tabletop exercises on whiteboards using the Emergo Train System[16] and one or two simulation exercises, such as traffic accidents. Other preparations varied from no other training (except for the above-mentioned 2-day training) (n=8) to disaster simulation exercises covering up to 4 hours (n=5), major incident training as part of being a voluntary staff member within a non-governmental organisation (n=2), academic courses on disaster medicine (n=1) or major incident exercises within the military services (n=1). None of the participants had participated in simulations covering more than 4 hours, neither in digital nor other kinds of simulation-based activities.

The patients' experiences in major incidents, as revealed in the interviews, included the following: a bridge collapse; car, bus or train accidents; a vehicle-ramming attack; an aircraft crash; a shooting; an explosion; a chemical incident and a fire in an elderly home care centre. Some participants had experienced several major incidents. According to the civilian post-traumatic checklist, none of the participants exhibited any signs of being traumatised by their experiences.

The following three main themes emerged: *skills and experiences are needed to manage major incidents, skills significantly covered in preparations* and *gaps in preparations in relation to the demands of real-life major incidents*. Six additional themes were identified: *manage the response, being in charge, frames for management, operational procedures, personal preparedness* and *professional experience* (see table 1).

### Skills and experiences needed to manage major incidents

This main theme describes the experiences of being a prehospital major incident commander in real-life events and the skills needed for such management.

### Manage the response

The first information presented from the dispatch centre to the incident commanders about the incident was vague, and they had limited information on the extent and character of the situation. Therefore, few of them were not mentally prepared to deal with a major incident when arriving at the scene.

To gain control of the situation and plan for dynamic development, the incident commanders strove to obtain an overview of the actual event as soon as they arrived at the scene. The importance of stopping and thinking, taking a breath and assimilating the scene instead of rushing into it was strongly emphasised:

> Stop and … and think. There is always time to stop and think about the whole thing before you get into such a situation. Even if it is chaotic when you arrive, it is worth taking these extra … to use 20–30 s and try to get a picture of the situation and the victims. You should try to make a plan before entering the scene. (Participant 5, referring to general experiences of being a prehospital major incident commander)

**Table 1** Overview of main themes, themes and subthemes

| Main themes | Skills and experiences needed to manage major incidents | | Skills significantly covered in preparations | Gaps in preparations in relation to the demands of real-life major incidents | | |
|---|---|---|---|---|---|---|
| Themes | Manage the response | Being in charge | Frames for management | Operational procedures | Personal processes | Lack of professional experience |
| Subthemes | To use a certain methodology | To feel lonely and exposed | Principles and methodology | Manage the complexity | Mental preparedness | Interpretation of the situation |
| | To grasp the situation | To be responsible for others | | Manage endurance | Ethical dilemmas | Mediating stress |
| | To make decisions | To manage stress | | | Self-knowledge | |
| | To improvise | To trust one's capability | | | | |

Several participants referred to a structure and methodology they had been taught to organise their response. Thus, sufficient knowledge of the principles and structure used to manage major incidents was necessary. Examples of such principles were mass casualty triage or reporting algorithms. Maintaining the agreed methodology, especially during the beginning of the response, was emphasised. However, details regarding the methodology and concepts differed from region to region, which is why local knowledge on, for example, when to report what and to whom, is needed.

Sensory impressions (such as vision, hearing, sight and smell), combined with professional experience and a 'gut feeling' based on knowledge and intuition, were used to interpret the situations and plan for consequences and possible developments. In this process, previous experiences from both real-life situations and training could contribute.

> I spontaneously felt in my body that I recognized this as a very difficult, major incident and at the same time I was somehow mentally prepared that something more would happen. So, I started to look for more injured. My intuition told me that there were more [injured] to find. (Participant 2, referring to a traffic accident)

Since conditions change every minute, the analysis of consequences and potential development of an incident was a continually ongoing process. Consequently, the development of these situations is hard to predict, meaning that incident commanders have to be constantly prepared to adapt their strategies and actions. Examples of unexpected conditions could be to find more victims, unexpected changes in patients' medical conditions, a sudden weather change or an escalating security situation. The incident commanders strive to be one step ahead and plan for different scenarios, using their professional experience and local knowledge to guide them:

> The conditions changed immediately at the triage site when one of the patients had convulsions. From being up walking, he just collapsed and developed general convulsions. I had to change the plan and made the helicopter take him immediately… (Participant 1, referring to a bridge collapse)

Several decisions had to be made, which in many cases had to be conducted quickly and based only on fragmented information. To make wise decisions, the incident commander requires good local knowledge (such as geography and logistical aspects). Knowledge of the capacity of nearby hospitals and personal knowledge of the ambulance personnel present at the scene facilitated the management and use of available resources. If local knowledge was lacking, the incident commanders risk making decisions with major consequences, such as making wrong or late decisions concerning the distribution of patients, possible locations for evacuation and other logistical questions.

### Being in charge

Being an incident commander entails the organisation of the medical response, leading colleagues and making decisions that would greatly impact others. The mandate of being an incident commander, which was both formal and informal, implies a greater power and responsibility compared with their normal work. To be trusted and respected, the incident commanders strive to clearly demonstrate their capability to manage the situation. For example, they use calm body language, talk slowly and loudly, and change their methods of communication to shorter and more concise information when compared with their everyday situations. Communication at the scene often relies on face-to-face communication, physical signals, such as a 'thumbs up', or eye contact. Sometimes, according to other colleagues, the incident commander does not fulfil their obligations. At such times, colleagues can feel frustrated, which in turn makes them feel obligated to make their own decisions to compensate for the perceived absence of management.

Most often, these decisions are time critical and could have life-changing consequences, which can cause added tension and ethical dilemmas. The decision-making was most often made by the incident commander on their own, and being aware that their decisions and actions could make the difference between life and death makes the incident commanders feel lonely and exposed:

> We had no one to ask. It was just my colleague and myself. There was no one else. Everything depended on us. We are used to it, but it's a thing when you have so many people that are dependent on you… I think maybe you do not understand this until after quite a few years within this profession. What you as incident commander decide can have huge consequences for others. We make decisions that otherwise the most senior medical doctor inside the hospital makes. You need to understand that the responsibility is huge. (Participant 12, referring to a bus–train crash)

Distanced support, such as from a remote senior advisor on commanding an incident, was not always perceived as supportive. If not present at the site, many of the impressions and complexities of situations may not be grasped, as it is difficult for someone not physically present to assimilate all the aspects of the situation. Technical means of communication (such as radios or mobile phones) are less supportive than expected, given the chaotic and often noisy environment. Another aspect of remote support was that it does not assist in the very early stages when many of the toughest decisions need to be made.

Maintaining the role of a manager also implies being responsible for ambulance personnel. In particular, security decisions such as if (and when) to send colleagues into collapsed buildings, rubble or hostile environments stressed the incident commanders:

> It is a dangerous world …. But if you have chosen to work within the ambulance services, then you need

to have courage, because we are the last outpost in society in some way. Although it can be scary, and it is dangerous, we must face it; we are not protected and if you choose this profession then you have to be prepared for that as well. I think it belongs to the profession. (Participant 9, referring to a terrorist attack)

It was emphasised that being among the ambulance personnel sometimes implies working in risky environments. However, a general lack of conformity regarding acceptable risks hampers the decision-making process and sometimes causes conflicts between the incident commander and ambulance personnel. The presence or perceived force in the stress reactions experiences by the prehospital major incident commanders, and how these reactions limit their cognitive ability, surprised the incident commanders. For example, the capability to maintain time perception was lost by many of the incident commanders. Stress does not only affect the incident commanders, but all personnel present at the site. Sometimes, the incident commanders must adapt their plans and their leadership style to the reactions exhibited by other ambulance or rescue personnel:

> To make wise decisions I must consider; do we have ambulance personnel on site who are in distress… so that they cannot handle it or… are my colleagues okay? That is also an important thing to consider when you make decisions. (Participant 10, referring to a bus crash)

A certain level of self-trust is necessary to enable the management of their own feelings and reactions while simultaneously managing the complexity of these situations. If the incident commander feels safe, trusted and supported by their colleagues, they can focus more on the actual incident management and less on their own reactions and limitations.

### Skills significantly covered in preparations

Some aspects were sufficiently covered in the preparations. The structure and principles, such as how to organise the response, nomenclature and reporting and triage systems, formed the basis for a structured methodology to manage major incidents:

> The moments that I really benefitted from were that you know that there is a structure, and you know who to contact and what to report when. (Participant 2, referring to traffic accident)

Being aware of the rules and regulations pertaining to the medical response and other stakeholders' mandates was necessary to enhance collaboration with those engaged stakeholders. Logistics, including the flow of patients and distribution of victims, are other issues that require specific knowledge. The sessions teaching such frameworks were mainly theoretical and were sometimes enacted in tabletop training. All these formal strategies

were sufficiently covered in the training, forming a basis for the incident commander to rely on.

### Gaps in preparations in relation to the demands of real-life major incidents

Within this theme, the gaps in preparations regarding the demands of managing a real-life event were described. The gaps found were related to operational procedures, personal processes and professional experiences.

#### Operational procedures

Real-life incidents are usually found to be more complex and harder to grasp than previous training sessions conveyed. Sensory impressions and timelines are especially hard to represent in simulations, and thus, training the use of intuition based on such impressions, or training stress reactions such as cognitive impairment, was difficult to obtain.

Collaboration with other actors at the scene (such as rescue services or the police) is necessary for the smooth management of the scene. This collaboration not only focuses on 'what to do' questions, such as methodological or strategic questions, but also on 'how to do it' questions, such as ethical considerations or leadership styles. These aspects were seldom discussed as part of the training sessions, causing a gap in the preparations. Most often, the training exercises focused on the first start-up phase, such as how to act when arriving at the scene. However, real-life major incidents last longer than any training or exercise.

> In general, I think we often talk about the start-up phase, about what we should do when arriving at the scene. But it is just as important to step down and end the mission in a good way. And all in between… to make sure that the right patient comes to the right hospital, that all colleagues are okay, that we have control of the equipment and so, to make sure that the step down is smooth…. We never train that. (Participant 13, referring to a bus crash)

The importance of preparing for the training of long-term management, stepping down and performing a closure of the site was emphasised.

#### Personal processes

While many processes on a personal level are activated during real major incidents, this did not occur during training or exercises. Sensory impressions, such as sights, sounds, smells or the general life and death awareness of actual victims compared with a manikin, were some of these. Such impressions increased the perceived stress in a way that was hard to attain in simulations:

> …but when you stand there, there are hearing impressions and you see things…and you smell things… and it is a warm body you hold in your hands and you feel your own pulse when you get stressed in a completely different way than during exercises. (Participant 11, referring to a terrorist attack)

Being mentally prepared is an essential part of the personal preparations required to hold the position of a prehospital major incident commander. Being mentally prepared mainly refers to a personal process of reflection on what it really means to be ambulance personnel in general and at a major incident in particular. Since the system implies that almost every ambulance personnel on duty could arrive at a major incident first and become the incident commander, this mental preparedness needs to be part of their everyday preparations when beginning their shift:

> You are never prepared when you come to work in the morning to be standing in the middle of a bridge collapse with 10 to 15 injured five hours later. You are not personally prepared for that. But as a professional, I need to be mentally prepared and think it though beforehand. I have to think, "How would I act if this happened?" That's your own responsibility and a good way to prepare yourself. (Participant 1, referring to general experiences from major incidents)

Being aware of one's own personal and professional qualities and limitations, and preparing to feel exposed, lonely and powerless, are also part of the mental preparation process. It was also stressed that preparing for such feelings involves listening and learning from others with experience of major incidents, in addition to using one's own imagination.

Self-knowledge, including being aware of how others perceive one's leadership and appearance, personal attributes and discussions on ethical dilemmas when managing real-life incidents are all highlighted as being absent from the training. Furthermore, the design and implementation of training affects personal preparations both positively and negatively. For instance, by acquiring a perceived feeling of having succeeded, rather than failing and learning from the mistakes, personal growth could be obtained during the training that could be used in real-life major incidents. By providing the chance to repeat one's performance until success was achieved, and by positive coaching and feedback during the training session, positive and increased self-esteem could be promoted. If the opportunity was provided to repeat the performance after feedback and in-depth reflections, a final feeling of having succeeded could be obtained. Such cognitive and physical memories could increase the possibility of acting correctly by instinct when faced with a similar challenge in the future.

### Lack of professional experience

Since major incidents are unusual events, many years of deployment within the ambulance service is required to gain sufficient experience to manage a major incident. In many situations, the personnel in managing positions at major incidents could be quite new and inexperienced. Professional experience from real-life major incidents could, to some extent, compensate for a lack of formal training and preparations. Having sufficient professional experience as ambulance personnel in an everyday perspective, specifically from acting as a prehospital medical commander in both minor and major incidents, is of great importance to be able to function in such a position during a major incident:

> You need experience, long experience. And you have to be able to manage much more than what they say on the course … that is not enough […] You need to take so much into consideration; geography, roles, laws and things like that. Then you need to be calm and confident in order to get people to act as you would like them to…so that they trust you. (Participant 8, referring to an explosion)

Professional experience adds value when analysing the potential development of a situation, when making decisions under pressure and for bearing the responsibility that comes with the role:

> Afterwards I can feel like "I fixed this," so I feel more self-confident […] I actually do the right thing and think the right things in situations like this. It was very positive for me; it has made me feel more confident in my professional role… (Participant 5, referring to a bus crash)

In addition, professional experience enriches self-trust and tolerance for one's own limitations and balances feelings of being exposed, which can make the incident commander more comfortable in the situation. Lacking professional experience has the opposite effect. Also, a lack of experience from major incidents and from working within ambulance services in general reduces the necessary ability of a commander in a major incident to tolerate and manage stress.

### DISCUSSION

This study identified areas that need further attention and applied guidance on how to prepare prehospital major incident commanders more effectively in the future. The conformity between real-life major incidents and preparations was good regarding sufficient preparedness for operational procedures such as organisation, reporting structure and triage methods. However, the preparations did not prepare for the complexity and endurance strategies required in real-life major incidents. Incident commanders were not prepared for several personal processes, such as stress management, ethical dilemmas and the self-knowledge required to handle real-life major incident situations. To some extent, professional experience could compensate for a lack of training. However, incident managers requested further, and to some extent different, preparations to incorporate operational procedures, professional experiences and personal preparedness to match the complexity and demands of real-life major incidents.

The incident commanders described how they used information from many sources, in addition to sensory

impressions and their own experience, to interpret the situation, predict development and make decisions. This process corresponds to the concept of situational awareness, as in the perception and understanding of the situation and its meaning.[17] Developing situational awareness is crucial for responding adequately to crises. However, this study demonstrated that this complex process was hard to grasp in the training and exercises, since these were not perceived as being as complex as reality. In particular, sensory impressions were hard to simulate, even though they have been found essential for assessing a major incident situation.[18] It is not known if simulated patients during training were used in the preparations or not, but such simulations might add more sensory impressions other than, for example, tabletop training. Additionally, the skill to improvise was not sufficiently covered in the preparations. Since improvisation is often used in emergency management,[17 19] its value needs to be further highlighted in the preparations.

The structure and method used for major incident management rely on a command and control paradigm aimed at controlling the situation. This type of methodology has long been considered the 'gold standard' for managing such incidents. However, this approach has been criticised for being unrealistic because it is seldom possible to control major incidents to such an extent, as the models suggest.[20] As found in this study, the assumption of being able to control a major incident in all aspects might cause doubts in one's capabilities, as well as feelings of being exposed and vulnerable if there is a failure to gain control of an incident. Such experiences have been previously described by other medical professionals when faced with disasters.[21–23] Therefore, it would appear important to reflect on how to balance the ability to control a situation and how to accept uncontrollable dimensions of a major incident in the preparations.

One question that caused a burden to the commanders pertained to the responsibility for the safety and security of colleagues. The same issue has been described among prehospital incident commanders during tunnel incidents[24] and terrorist attacks.[18] During the interviews, reflections about the acceptable levels of risks, formal and moral obligations, legal and professional responsibilities of ambulance personnel in general were raised, as well as dilemmas in the specific context of a major incident. It became clear that these discussions were an important part of one's professional identity and mental preparedness for managing major incidents.

Managing stress was another challenge raised by the participants. Since one of the characteristics of major incidents is a high level of uncertainty,[2 3] and the essence of stress has been suggested to consist of uncertainty,[25] this finding is not surprising. The job demand-control-support model[26] that has been widely used to explain work-related stress illustrates the relation between demands, both quantitative, qualitative, and emotional, in relation to experience level of control and available support. Acting in an unfamiliar position, being witness to a large amount of human suffering and managing safety and security issues and high expectations both from oneself and others were experiences expressed by the prehospital major incident commanders. The combination of high demand and low control may lead to an increased risk of both physical and psychosocial health problems in both a short-term and long-term perspective.[27] It has been suggested that incident managers' personnel need to develop strong self-resilient strategies to manage the stress inherent in both everyday work and critical incidents.[28] However, stress management and the resources needed to stay resilient during and after stressful and potentially traumatic events are not only a personal choice, but these are highly dependent on organisational support.[29] This statement is supported by the finding that the social support from colleagues and managers is an important 'buffer' that may moderate the negative effects on perceived stress, mental health and job strain.[27] Therefore, individual stress management strategies as well as a supportive climate within the working group should be seen as an essential preparation among prehospital major incident commanders to enhance a resilient response. Also, mental preparedness including ethical and moral reflections seems to be an essential component to minimise the gap between experiences and formal demands and experienced level of control. However, the health consequences from being a prehospital incident commander and their relation to preparations are areas that remain unexplored, requiring further research interest.

Professional experience was found to mediate some of the gaps between preparations and real-life experiences. Since few incident commanders will have the possibility of gaining rich experience from real-life major incidents, training and other kinds of preparations are needed. The evaluation and feedback to participants in major incident training are often informal[30] or, as stated by the participants, focused on errors and shortcomings. By focusing on positive post-exercise debriefings[29] and by stimulating self-reflection during training,[31] professional experience and self-confidence can be better enhanced. In some organisations, video cameras or drone cameras are being used to document the scene and provide an overview of the situation. Such tools may add value to learning debriefings. In addition, it can be suggested that learning debriefings or other feedback sessions focus not only on technical procedures and skills, but also on interpersonal and personal competences, such as leadership, communication skills and stress management. As suggested by the participants in this study, sharing experiences from real major incidents can also enrich personal preparations and add several dimensions except from the technical or organisational aspects usually discussed in lesson-learnt sessions. Also, new technology and new simulations can open the door to new possibilities to train and prepare for a wider array of conditions than the traditional major incident and disaster exercises.[32] Since the knowledge on what particular exercise components that are vital for

achieving learning in major incident or disaster training for healthcare professionals is still limited,[33] this is a field requiring more scientific interest.

## Limitations

As in all qualitative studies, the results of this study rely on a sample of personal experiences. The two-session approach[13] was found to stimulate a deeper reflection and provided rich material to analyse. The second interview sessions were used to follow up on the first ones and were therefore shorter. Some participants did not choose to participate in the second interview, but they agreed to be included in the study based on their first interview session. Another limitation was the lack of detailed information collected on the participants' preparations in terms of exact content of courses or training. However, this qualitative study does not aim to distinguish between specific training or preparations. The discussion during the analysis between all three authors provided a broad basis for the analysis, enriched by different professional backgrounds covering, among other things, disaster medicine, crises management, health professions education and professional development.

## CONCLUSION

This study identified perceived gaps between preparations and real-life experiences of being a prehospital major incident commander. To minimise these gaps between demands and expectations on perceived control, and better prepare individuals for being prehospital major incident commanders, the training and other preparations should reflect complexities of real-life incidents. Preparations should promote and develop both technical skills required, such as principles and methodology used in major incidents, and personal preparedness. Personal preparations should include improving one's mental preparedness, self-knowledge and professional self-confidence required to successfully act as a prehospital incident commander. However, the design and conduct of effective educational sessions and training exercises and what pedagogical methods should be used to develop both technical skills required and build the necessary professional and personal preparedness is still a challenge that needs to be further investigated. In addition, studies on the relation between preparations and health consequences from being a prehospital major incident commander are areas requiring further attention.

**Contributors** All authors participated in the planning and design of the study. KH conducted all interviews and analysed the data, during discussions with SE and KB. KH was the major contributor in writing the manuscript, with the assistance of SE and KB. All authors read and approved the final manuscript.

**Funding** The authors have not declared a specific grant for this research from any funding agency in the public, commercial or not-for-profit sectors.

**Competing interests** None declared.

**Patient and public involvement** Patients and/or the public were not involved in the design, or conduct, or reporting, or dissemination plans of this research.

**Patient consent for publication** Not required.

**Ethics approval** The study was reviewed and approved by the Swedish Ethical Review Authority (ref id: 2018:232).

**Provenance and peer review** Not commissioned; externally peer reviewed.

**Data availability statement** No data are available. The data analysed during the current study are not publicly available due to Swedish laws on research ethics and the ethical approval.

**ORCID iD**
Karin Hugelius http://orcid.org/0000-0003-0534-4593

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
