## [Reviewer comments · BMJ Open]

ARTICLE DETAILS

TITLE (PROVISIONAL)	Prehospital major incident management: How do training and real-life situations relate? A qualitative study.
AUTHORS	Hugelius, Karin; Edelbring, Samuel; Blomberg, Karin

VERSION 1 – REVIEW

REVIEWER	Gyllencreutz, Lina Department of Nursing, Umeå University
REVIEW RETURNED	07-Mar-2021

GENERAL COMMENTS	I have reviewed the manuscript Prehospital major incident management- how does training and authentic situations relate? I think that the authors address a very interesting and relevant topic and contribute with valuable knowledge within this area. Thank you. Below you find my comments for each section of the manuscript. Thank you for the opportunity to review the manuscript. Abstract The objective at page 2 and page 3 differ from each other. For convenience use the same wording. The methods in the abstract are not convenience with what is said in the manuscript. For example, in the abstract the interviews were done within one month but in the manuscript you say that it was done within two weeks. Manuscript Methods. Page 3 line 50 the concept “first responders” is used. Previous and later in the manuscript you use ambulance personnel. Data collection procedure. The two-session interview approach is interesting but would benefit of being a little bit more explained. Please refer to the reference you used. Why did you choose the interval of two weeks? Why did you choose to include the three participants that did not complete the second interview? It would be interesting to know what kind of questions you asked during the first interview session and if you asked the same type of questions during the second session or if that session contained a different type of question? The second interview session lasted shorter than the first session. How come? Did you expect that? In addition, the Posttraumatic checklist was used. I didn't see that coming. Why did you choose to assess the participant's post-
--

	traumatic stress level? Please provide a better background description about why you choose to incorporate PTSD. Results. I lack information about the participant's preparations. Please provide information about what kind of exercise/training the participants have received? To me, the main themes seem to be the same as the two issues in the objective? Overall, it is difficult to understand how the themes and subthemes relate to the objective especially when it comes to the relationship between the preparation and authentic incidents. For example, the subtheme 'manage endurance'. For me, that subtheme indicate that the preparation includes to be able to manage endurance but the result says that the participants lack that kind of preparation. I encourage the authors to re-review the themes and subthemes based on how the themes and subthemes respond to the objective. Generally, I think that the result would benefit of being described the other way around i.e. start with a description of what kind of training/exercise the participants have and then how that relate/or not to authentic incidents. It seems preferable to first be prepared and then execute the skills in an authentic incident. Page 8 line 23: what do you mean with one more theme was included with the main theme preparations? I don't understand how a theme could be placed between the two main themes? Please be a little bit clearer in the method section. Figure 1. Decision is misspelled.
--	--

REVIEWER	Alinier, Guillaume Hamad Medical Corporation, Ambulance Service
REVIEW RETURNED	15-Mar-2021

GENERAL COMMENTS	Thank you for the opportunity to review your article submitted to BMJ Open. As this is an activity in which I am engaged in, I read it with great interest. The article is relatively well written (minor typos) and is well structured and informative. The review process would have been a bit easier if you had used continuous line numbering. Please remember for the next submission. Abstract: The abstract says very little about the type of "preparations" the authentic events are compared to, so it is not very informative. Are the preparations simply lectures, computer-based exercises, or large scale simulations? At present the readers cannot make any judgement and recommendations in the conclusion remain vague as to what educational strategy/methods should be adopted. L31: Missing word "...management was sufficiently..." Article: Background, L10: The definition you provide here is that of a "crisis"! Please review. L14: "decision-making" (typo). L19: What do you mean by "operate all ambulances"? L32: Typo in "professional experience". It was a good that you subjected participants to the post-traumatic checklist! Good description of the analysis. Table 1, L46: Should be "one's" instead of "once". Page 5, L57: Please clarify information presented by who? From the dispatcher in the call-centre?
--

	P6, L17-21: Can you cite some of the methods used? I am a bit surprised that your article does not refer to the use of the METHANE report at this point. P6, L55-58: Could be better phrased, simplified. "... making wrong decisions concerning the distribution of patients..." Several times through the article you refer to "training and exercises" which is somehow an expression I find strange. Why not "training exercises" or "educational sessions and training exercises"? It seems like the staff you interviewed have never taken part in large scale major incident simulation exercises and multiple agencies and hospitals over a whole day or night! Expensive and resource intensive to run, but great learning for everyone at all levels! The learning is during the event and during the debriefings that occur later at different levels, and the reports produced following the event, with the learning points. A section describing their experiences in terms of training would have been useful, like you briefly described the types of real major incidents they have attended and if some had attended several major incidents. P5, L29: Where is Table 1? P9, L12: Something is not right in that sentence. Maybe it should be "... of training for long term management..." P9, L20-21: The end of that sentence needs rephrasing. The comments in the quotations show that the participants did not get exposed to highly immersive and realistic major incidents simulations with simulated patients with moulage (make up for wounds) for example, which should be a recommendation from your article. P9, L40: Should be "personally". P9, L42: Should it be "through" instead of "though"? P9, L58: Good point, but could not apply to huge full scale major incident exercises... P10, L34: "I am more confident in myself" or "I am more self-confident". (not both) P10, L55: Clearly applies to your sample of participants and should maybe be specified as some other services already address these points. P11, L35: "previously". Use the full term "debriefings" instead of "debriefs" throughout. P12, good section L9-21. References: It is limited in this specific domain, but I would recommend referring to more articles pertaining to simulation-based training and debriefing for major incident exercises.
--	---

VERSION 1 – AUTHOR RESPONSE

Reviewer: 1

Dr. Lina Gyllencreutz, Department of Nursing, Umeå University Comments to the Author:

I have reviewed the manuscript Prehospital major incident management- how does training and authentic situations relate? I think that the authors address a very interesting and relevant topic and contribute with valuable knowledge within this area. Thank you.

Below you find my comments for each section of the manuscript. Thank you for the opportunity to review the manuscript.

Reply: Thank you for the positive words, and for providing useful comments and suggestions to improve the quality of this paper. We appreciate your efforts very much!

Abstract

The objective at page 2 and page 3 differ from each other. For convenience use the same wording. The methods in the abstract are not convenience with what is said in the manuscript. For example, in the abstract the interviews were done within one month but in the manuscript you say that it was done within two weeks.

Reply; Thank you for making us aware of this. The aim has been revised and is now the same in abstract and text. Also, suggested revision of the method section in the abstract has been made.

Manuscript

Methods. Page 3 line 50 the concept "first responders" is used. Previous and later in the manuscript you use ambulance personnel.

Reply; Thank you for notifying this. The wording has been changed to ambulance personnel.

Data collection procedure. The two-session interview approach is interesting but would benefit of being a little bit more explained. Please refer to the reference you used. Why did you choose the interval of two weeks? Why did you choose to include the three participants that did not complete the second interview? It would be interesting to know what kind of questions you asked during the first interview session and if you asked the same type of questions during the second session or if that session contained a different type of question? The second interview session lasted shorter than the first session. How come? Did you expect that?

Reply; As stated in the manuscript, we really found the two- session interview approach helpful to make sure that all thoughts or experiences, also those emerging after the first interview, were included in the data. Furthermore, we link the method stronger to the literature by adding the reference Read (2018) from which we draw on in our process (methods section). We have added text describing the questions used in the second interview ("In the second interview, the first interview was followed up on, and the study participants was asked if he or she had any additional thoughts or experiences that had not been mentioned in the first interview.") to clarify the difference between the two sessions. Depending on that the second interview was more of a follow up, these were in general shorter than the first session interview. This shas been added in the method discussion section (The second interview session was used to follow up on the first interview and was therefore in general shorter than the first one. Some participants did not choose to participate in the second interview but agreed to be included in the study based on their first interview session.)

In addition, the Posttraumatic checklist was used. I didn't see that coming. Why did you choose to assess the participant's post-traumatic stress level? Please provide a better background description about why you choose to incorporate PTSD.

Reply: Additional data gained from the interviews in this study will be used to report on stress and recovery strategies after being deployed in prehospital major incidents (that study will be reported elsewhere). Therefore, the Posttraumatic checklist-civilian was therefore used to scan for potential indications of PTSD (that will be further reported in a separate study) and also to ensure the wellbeing of the study participants, in accordance with the approval from the Swedish Ethical Review Authority. Even if not used as part of the result in this study, we choose to report the use and results of the scan also in this paper, in order to be transparent.

Results. I lack information about the participant's preparations. Please provide information about what kind of exercise/training the participants have received?

Reply: We agree that this information is necessary in order to interpret the result and has added information on the preparations of the participants in the results.

To me, the main themes seem to be the same as the two issues in the objective? Overall, it is difficult to understand how the themes and subthemes relate to the objective especially when it comes to the relationship between the preparation and authentic incidents. For example, the subtheme 'manage endurance'. For me, that subtheme indicate that the preparation includes to be able to manage endurance but the result says that the participants lack that kind of preparation. I encourage the authors to re-review the themes and subthemes based on how the themes and subthemes respond to

the objective.

Reply: Thank you for this suggestion. We have revised the result presentation and agree that the previous presentations do not really reply to the objective. Therefore, we have revised the presentation of the result, to more clearly address the gaps found. A revised Table 1 has been added and the Figure 1 has been removed, since we think the Table 1 clearly describe the results. We hope that this better illustrate the results. Also, the text on endurance has been revised.

Generally, I think that the result would benefit of being described the other way around i.e. start with a description of what kind of training/exercise the participants have and then how that relate/or not to authentic incidents. It seems preferable to first be prepared and then execute the skills in an authentic incident.

Reply: We understand the point in this suggestion and have considered this approach. However, the study focuses on the gaps, and the demands in real events, why we think it is valid and more correct to present these first, and then relate these to the preparation and training. This has been further clarified in the analysis description.

Page 8 line 23: what do you mean with one more theme was included with the main theme preparations? I don't understand how a theme could be placed between the two main themes? Please be a little bit clearer in the method section.

Reply: We agree that this statement confuses the result presentation, and we have therefore removed it.

Figure 1. Decision is misspelled.

Reply: Thank you for notifying us on this.

Reviewer: 2

Dr. Guillaume Alinier, Hamad Medical Corporation, University of Hertfordshire Comments to the Author:

Thank you for the opportunity to review your article submitted to BMJ Open. As this is an activity in which I am engaged in, I read it with great interest. The article is relatively well written (minor typos) and is well structured and informative.

The review process would have been a bit easier if you had used continuous line numbering. Please remember for the next submission.

Reply: Thank you very much for the encouraging words. We apologize for the numbering mistake.

Abstract:

The abstract says very little about the type of "preparations" the authentic events are compared to, so it is not very informative. Are the preparations simply lectures, computer-based exercises, or large scale simulations? At present the readers cannot make any judgement and recommendations in the conclusion remain vague as to what educational strategy/methods should be adopted.

Reply: Thank you for addressing this problem. We have revised the abstract and added information on the study person's preparations and authentic experiences.

L31: Missing word "...management was sufficiently..."

Reply: Thank you, this has been corrected.

Article:

Background, L10: The definition you provide here is that of a "crisis"! Please review.

Reply: Thank you for this suggestion. We have reviewed the definition in the background against the references again and consider them to be valid.

L14: "decision-making" (typo).

Reply: Thank you, this has been corrected.

L19: What do you mean by "operate all ambulances"?

Reply: The sentence addresses the fact that in Sweden, at least one nurse, often specialized in prehospital emergency care, is on duty in each emergency ambulance. We have revised the writing to "is on duty in all emergency ambulances" to clarify.

L32: Typo in "professional experience".

Reply: Thank you, this has been corrected.

It was a good that you subjected participants to the post-traumatic checklist!
Good description of the analysis.

Reply: Thank you!

Table 1, L46: Should be “one’s” instead of “once”.

Reply: Thank you, this has been corrected.

Page 5, L57: Please clarify information presented by who? From the dispatcher in the call-centre?

Reply: Thank you for addressing this. The sentence has been revised to: The first information presented from the dispatch centre to the incident commanders about the major incident was vague.

P6, L17-21: Can you cite some of the methods used? I am a bit surprised that your article does not refer to the use of the METHANE report at this point.

Reply: The METHANE report is included in the MIMMS methodology, and also part of a concept used for communication and reporting in major incidents in Sweden. This concept is named Prehospital Major Incident Command concept, and is adopted by many, but not all, Swedish ambulance services and there are several versions of the concept within the country. Some examples of methods used and a clarification of local variations has been added to the results.

P6, L55-58: Could be better phrased, simplified. “... making wrong decisions concerning the distribution of patients...”

Reply: Thank you, this has been corrected.

Several times through the article you refer to “training and exercises” which is somehow an expression I find strange. Why not “training exercises” or “educational sessions and training exercises”?

Reply: Thank you for this supportive suggestion. The phrasing has been adjusted throughout the manuscript.

It seems like the staff you interviewed have never taken part in large scale major incident simulation exercises and multiple agencies and hospitals over a whole day or night! Expensive and resource intensive to run, but great learning for everyone at all levels! The learning is during the event and during the debriefings that occur later at different levels, and the reports produced following the event, with the learning points.

A section describing their experiences in terms of training would have been useful, like you briefly described the types of real major incidents they have attended and if some had attended several major incidents.

Reply: We strongly agree that it is surprising that no participant had been part of a longer exercise. We have added information on the study participants’ experiences of real major incidents, type of incidents and also their preparations, such as educational sessions and training exercises.

P5, L29: Where is Table 1?

Table 1 is included in the manuscript, in the beginning of the result.

P9, L12: Something is not right in that sentence. Maybe it should be “... of training for long term management...”

Reply: Thank you, this has been corrected.

P9, L20-21: The end of that sentence needs rephrasing.

Reply: Thank you, this has been corrected.

The comments in the quotations show that the participants did not get exposed to highly immersive and realistic major incidents simulations with simulated patients with moulage (make up for wounds) for example, which should be a recommendation from your article.

Reply: Thank you, this is a very good point, and we have added it to the recommendations.

P9, L40: Should be “personally”.

Reply: Thank you, this has been corrected.

P9, L42: Should it be “through” instead of “though”?

Reply: Thank you, this has been corrected.

P10, L34: “I am more confident in myself” or “I am more self-confident”. (not both)

Reply: Thank you, this has been corrected.

P11, L35: “previously”.

Reply: Thank you, this has been corrected.

Use the full term “debriefings” instead of “debriefs” throughout.

Reply: Thank you, this has been corrected.

P12, good section L9-21.

References: It is limited in this specific domain, but I would recommend referring to more articles pertaining to simulation-based training and debriefing for major incident exercises.

Reply: Thank you for suggesting this. We have reviewed the literature again, and reference 1, 11 and 26, 27 and also inserted ref 28 related to this topic.

VERSION 2 – REVIEW

REVIEWER	Alinier, Guillaume Hamad Medical Corporation, Ambulance Service
REVIEW RETURNED	26-Jun-2021

GENERAL COMMENTS	Thank you for the opportunity to review your revised article submitted to BMJ Open. As mentioned in my first review, using continuous line numbering would have made providing feedback easier/quicker. Please remember for the next submission. I notice that many of the points I have raised have not been addressed. Your article should be totally proof read and edited by a native English speaker as much of the terminology is not accurate and makes the article difficult to read. Abstract: Objective: what do you mean by “authentic experiences”, is that “actual/real life” ones or “realistic” ones? (Apply the change throughout the article) Change word “acting” as it is confusing in the training versus real context. It can simply be “...among prehospital major incident commanders.” The methods’ sentence needs to be rewritten. Last sentence of conclusion sounds repetitive. Here and in the results section, what do you mean by “personal preparations”? Article: Background, L18: The definition you provide here is that of a “crisis”! Please review. L22: “decision-making“ (typo). L27: should be “... in each emergency ambulance.” L39: Replace “comparatively” by “relatively”. L40: Typo in “professional experience”. L46 and 54: would read better as “prehospital major incident commanders”. L59: Do you mean “exposure”, or have the singular form of “preparation”. P4L5: Remove term “acting” and reword as “prehospital major incident commander”. Why did you switch to “manager” here? Throughout, adopt “held the role” instead of “acted/acting”. P4L19: “...participants to provide further information about the study...” P6L27: Please rephrase as per earlier feedback. P8L4: You have used many quotations from your interviewees, however there is no context. In the bracket where you put the participant number, please add the type of incident to which the comment referred as it would help give the quote some context.
---

	P8L12: Needs to be rephrased P8L17: Many rescue services now have video systems on major incident command vehicles and drones. This aspect should be part of your discussion. P9L18: Replace “was” by “were”. P9L24: Needs to be rephrased. P9L45: Needs to be rephrased. P10L3: There was an issue of “fidelity” of the exercises/simulations, which needs to be discussed later. P10L6-14: Needs to be reworded to avoid repetitions. Several typos in the quote that follows. P10L44-50: Needs to be revised. Discussion: Limited depth and scope of discussion especially as very limited information as to the type of training and level of reality of the simulation exercises the staff were involved in. Not having collected more information about this is a major limitation of the present study and renders the collected feedback not very useful. More literature around the topic should be explored and cited. This seriously limit the academic quality of the current study. Conclusion: Too limited
--	--

VERSION 2 – AUTHOR RESPONSE

Reviewer: 2

Dr. Guillaume Alinier, Hamad Medical Corporation, University of Hertfordshire Comments to the Author:

Thank you for the opportunity to review your revised article submitted to BMJ Open. As mentioned in my first review, using continuous line numbering would have made providing feedback easier/quicker. Please remember for the next submission. I notice that many of the points I have raised have not been addressed. Your article should be totally proof read and edited by a native English speaker as much of the terminology is not accurate and makes the article difficult to read.

Reply: Thank you for taking your time and efforts to improve our manuscript. The revised manuscript has now been further edited for English language by a professional academic editing service, and we hope that this improves readability of the paper. In accordance with journal instructions for authors, no line numbers has been added, since the submitting system adds line numbering. Therefore, we hope that you will be able to use those.

Abstract:

Objective: what do you mean by “authentic experiences”, is that “actual/real life” ones or “realistic” ones? (Apply the change throughout the article) Change word “acting” as it is confusing in the training versus real context. It can simply be “...among prehospital major incident commanders.”

Reply: Thank you for this suggestion. We agree and have discussed within the research team and agreed on to consistently use the term real-life situations/experiences/ events.

The methods’ sentence needs to be rewritten.

Reply: The section has now been further review for English language by a professional editing services, and we have made changes in accordance with their suggestions.

Last sentence of conclusion sounds repetitive. Here and in the results section, what do you mean by “personal preparations”?

Reply: Thank you for addressing this. We have revised the conclusion section.

Article:

Background, L18: The definition you provide here is that of a “crisis”! Please review.

Reply: Thank you for this observation. However, we have used a definition found in the reference 1 paper (first line in the introduction in that paper) in which the interested reader can find further clarification and therefore, we have stayed with that definition.

L22: "decision-making" (typo).

L27: should be "... in each emergency ambulance."

L39: Replace "comparatively" by "relatively".

L40: Typo in "professional experience".

L46 and 54: would read better as "prehospital major incident commanders".

L59: Do you mean "exposure", or have the singular form of "preparation".

P4L5: Remove term "acting" and reword as "prehospital major incident commander". Why did you switch to "manager" here? Throughout, adopt "held the role" instead of "acted/acting".

P4L19: "...participants to provide further information about the study..."

P6L27: Please rephrase as per earlier feedback.

Reply: Thank you for the above suggestions and observations. We have revised all in accordance with your suggestion.

P8L4: You have used many quotations from your interviewees, however there is no context. In the bracket where you put the participant number, please add the type of incident to which the comment referred as it would help give the quote some context.

Reply: Thank you for this suggestion, we have added information on the context of the quotations.

P8L12: Needs to be rephrased

Reply: Thank you for this observation, we have revised.

P8L17: Many rescue services now have video systems on major incident command vehicles and drones. This aspect should be part of your discussion.

Reply: Thank you for this suggestion, we have added this perspective in the discussion.

P9L18: Replace "was" by "were".

P9L24: Needs to be rephrased.

Reply: Thank you for this observation, we have revised.

P9L45: Needs to be rephrased.

Reply: Thank you for this observation, we have revised.

P10L3: There was an issue of "fidelity" of the exercises/simulations, which needs to be discussed later.

Reply: Thank you for raising this. We are not quite sure of what you mean, and therefore, we have not been able to discuss this aspect.

P10L6-14: Needs to be reworded to avoid repetitions. Several typos in the quote that follows.

Reply: Thank you for this observation, we have revised.

P10L44-50: Needs to be revised.

Discussion: Limited depth and scope of discussion especially as very limited information as to the type of training and level of reality of the simulation exercises the staff were involved in. Not having collected more information about this is a major limitation of the present study and renders the collected feedback not very useful. More literature around the topic should be explored and cited. This seriously limit the academic quality of the current study.

Reply: We agree that it is a limitation that no deeper information on the specific trainings and preparations for the participants can be provided. However, we have added information on the content and forms of the ordinary two-day training for prehospital major incident commanders, that most participants had obtained. To deeper the discussion and increase the academic quality of the discussion, we have used the theoretical model of demands, control, support model to discuss our findings, and related them to preparations for being a prehospital major incident commander.

Conclusion: Too limited

Reply: Thank you for this comment. After discussion within the research team, guided by the authors instructions, we have revised the conclusion section.